# Formation of Ultimate Thin 2D Crystal of Pt in the Presence of Hexamethylenetetramine

**DOI:** 10.3390/ijms231810239

**Published:** 2022-09-06

**Authors:** Siti Naqiyah Sadikin, Marjoni Imamora Ali Umar, Azrul Azlan Hamzah, Muhammad Nurdin, Akrajas Ali Umar

**Affiliations:** 1Institute of Microengineering and Nanoelectronics, Universiti Kebangsaan Malaysia, Bangi 43600, Malaysia; 2Department of Physical Education, Faculty of Tarbiyah, Insitut Agama Islam Negeri Batusangkar, Batusangkar 27213, Indonesia; 3Department of Chemistry, Faculty of Mathematics and Natural Sciences, Universitas Halu Oleo, Kendari 93132, Indonesia

**Keywords:** 2D crystals, Pt, fcc metal, ultra-thin sheet, combinative effect

## Abstract

Platinum naturally crystalizes into a three-dimensional crystal due to its highly symmetrical fcc lattice, with a metallic bond which is non-directional and highly isotropic. This inherently means ultimately that 2D crystals of a few atoms thick growth are hardly available in this material. Here, we discovered that a combinative effect of formic acid reductant and hexamethylenetetramine surfactant during the reduction of their metal ions precursor can realize an ultimate thin 2D crystal growth in platinum. High-resolution transmission electron microscopy and filed-emission electron microscopy analysis have also discovered that the 2D crystal of Pt has 111 facets with a lateral dimension that can be up to more than 5 μm × 2 μm. The thickness of the 2D crystal of Pt is 1.55 nm. A mechanism for obtaining ultimate thin 2D crystal of Pt using the present approach is proposed.

## 1. Introduction

Atoms’ thick two-dimensional (2D) crystal, particularly graphene-like lattice crystals, have been the focus of attention in the material chemistry synthetic field for the past decade [1]. With a structure a few atoms thick, this 2D crystal allows an ultimate 2D confinement of the electronic carriers, producing many new phenomena, such as enhancement of exciton formation and robustness [2], in-plane plasmonic resonance and optical nonlinearity [3], strong light-mater coupling [4], thickness and dimension dependent properties [5,6], as well as ultra-fast energy-transfer at the surface [7]. Owing to such interesting phenomena arising from the 2D crystal system, attempts towards the transformation of other technologically important materials, such as platinum, into such an ultimate thin 2D system should be continuously attempted for enhanced performance in scientific and industrial applications. 

Platinum (Pt) nanostructures are known as efficient catalysts in many reactions that range from oxygen and carbon monoxide reduction reaction [8] to fuel cells [9], hydrogenation or hydroxylation [10,11], pollutant reduction [12,13], hydrogen storage [14], and a wide range of organic reactions [15]. An effort to enhance their performance in existing applications has been continuously carried out by preparing them in the form of variable shapes [16,17] and in the form of different crystalline phases [18]. The application of single-atom Pt as a catalyst has also been reported recently [19,20,21]. Owing to its unique 2D quantum confinement effect, the ultimate 2D crystal of Pt may produce exceptionally high performance in existing applications. Therefore, extensive attempts should be continuously demonstrated to realize such atoms’ thick 2D crystal growth in this material system. However, contradictory to this emerging interest, because of the nature of the highly isotropic character of the metallic bond in Pt, the nanocrystal favors growing into a spherical morphology and ultra-thin 2D crystal growth, to our best knowledge, has never been realized so far in this material. Nevertheless, there have been attempts to mimic the 2D structure of the Pt by assembling its colloidal crystal via the Langmuir-Blodgett method [22], but the unique properties resulting from the 2D quantum system, as in the single-crystalline system, cannot be obtained from this 2D colloidal assembly. Despite a recent study indicating that thin Pt nanoplates can be realized using templated epitaxial growth of Pt nanostructure on various crystalline substrates [23,24] and although the thickness and dimension of the nanoplate can be reduced down to a few atoms, the existence of crystalline substrate strongly influences the properties of the 2D crystal of Pt. Thus, the free-standing ultimate 2D crystal of Pt should be realized.

Here, we discovered that the ultimate thin 2D crystal of Pt can be realized using a simple aqueous phase reduction of Pt precursors by formic acid in the presence of hexamethylenetetramine surfactant. Although the lateral dimension of the 2D crystal of Pt is still relatively small, i.e. 5 μm × 2 μm, they may generate peculiar physico-chemical properties for enhanced performance in existing applications. 

## 2. Results and Discussion

Figure 1 shows the typical 2D crystal of Pt that is produced in this approach. The dimension of the 2D crystal is several tens of square nanometers. Zoom-in imaging at the position of c in Figure 1a indicates a detailed lattice of 2D crystals (Figure 1c–e). It is indicated that this 2D crystal lattice has a hexagonal geometry, which could be related to the (111) facet of the Pt. As also seen in a, the 2D crystal is very thin so it exhibits a similar electron contrast with the amorphous carbon film background. Selected area diffraction analysis of the lattice confirms its hexagonal geometry (Figure 1f). Bright halos in the background of the SAED pattern originate from an amorphous carbon film support layer. As also can be seen from the SAED pattern, the six white spots have the same brightness. This can only be obtained if the elements that construct the hexagonal lattice lie at the same level in the lattice. In other words, they must be in a common lattice plane. This, therefore, reveals that the localized 2D crystal is a perfect ultra-thin sheet with a very smooth and flat structure. Using the typical TEM image of the sample of area 200 nm × 200 nm, we estimated the yield of the 2D crystal of Pt. We found a 2D crystal share, approximately, as high as 40% of the product, with the rest belonging to thin and irregular morphology nanostructures (Figure 2).

To verify the element that constructs the 2D crystal, we carried out an Energy-Dispersive X-ray elemental analysis (Figure 3a,b). As the EDX spectrum reveals, the thin sheet structure on the Cu grid, which is the structure that is identical with those found on the HRTEM image, is composed of a Pt element. The presence of Cu and C elements in the spectrum is also recorded. However, originates from the TEM grid substrate background. Thus, this result confirmed the formation of 2D Pt crystals.

We then performed filed-emission electron microscopy analysis to further verify the existence of the 2D crystal of Pt, along with its dimension. The 2D crystal of Pt has a dimension up to 5 μm × 2 μm (Figure 4a). We also found that the 2D crystals tends to stack onto each other on the substrate. This could be the result of minimizing the high-surface energy of the 2D crystal system. A similar fact is also observed in the HRTEM analysis result and the atomic force microscopy (AFM) analysis results (which will be discussed in the following section). Again, EDX analysis during the FESEM analysis further confirms that the 2D crystal’s system is Pt (see inset in Figure 4a). Figure 4b–d show the confocal microscope (Figure 4b) and AFM image (Figure 4c,d) that was taken at the red arrow position in Figure 4b. As mentioned above, the 2D layer sample is composed by stacking several 2D crystals of Pt. Line profile analysis on the AFM image (Figure 4d) indicated that the thickness of a single layer of the 2D crystal of Pt is approximately 1.55 nm. This is an extremely thin Pt nanostructure that may produce peculiar physicochemical properties for high performance in existing applications. As shown in the inset of Figure 4d, the edge of the stacked 2D crystal is likely to be tilted up. Nevertheless, the definite structure and dimensionality of the 2D crystal of Pt might be further obtained by applying a proper processing technique for the AFM data. This includes the elimination of the substrate inclination relative to the AFM probe so the accuracy of the layered structure thickness or spacing can be increased. Despite this fact, the thickness of the layer can be roughly estimated from the result. In addition, a more detailed confocal microscope image of the 2D crystal of Pt might also be obtained via differential interference contrast (DIC) microscopy or Nomaski’s microscopy [25,26]. However, although Nomaski’s microscopy analysis is not available at present, the current result more or less verifies the existence of the ultimate thin 2D crystal of Pt.

We carried out a low-angle X-ray diffraction analysis on the samples to verify the phase crystallinity of the sample. A grazing angle as low as 1.8°, a small scan window, and a scan speed of 10°/min were used during this analysis. This condition was used due to in normal ultra-thin nanosheet of fcc metal [27], and the diffraction from the higher index plane is hardly detected. The result is shown in Figure 5. As can be seen from Figure 5, a strong diffraction peak is observed at 2θ of 38°. This peak can be annotated as (111) Bragg plane of the fcc Pt [28] (JCPDS file 70.2057). Thus, this result verifies the phase crystallinity of the 2D crystal system. 

The reduction of Pt^4+^ by the formic acid in the presence of HMT follow the formulas given in Equations (1) and (2) [29].
(1)$$K_{2}{\mathrm{PtCl}}_{6}{+H}_{2}O+\mathrm{HCOOH}\to {\mathrm{Pt}+\mathrm{CO}}_{2}↑{+\mathrm{KCl}+H}_{2}↑$$
(2)Pt+HMT→Pt·HMT

Metallic Pt along with KCl and gaseous CO_2_ and H_2_ was produced in this reaction. However, the formation mechanism of the 2D crystal of Pt using the present method is not yet understood currently. We hypothesize that the 2D crystal of Pt is realized via a template-assisted growth process, i.e., a two-dimensional molecular scaffold that is formed from the interaction of formic acid dimers in the reaction via π-π stacking interaction. It is well known that the formic acid molecule is stable in the form of the hexagonal cyclic dimer [30,31] and has dual functionalities as an acid and as an aldehyde [32], functioning as the reducing agent for the metallic cations and surfactant, respectively. We thought that HMT with a tetrahedral cage structure may in the first instance act as a host [33,34] for the as reduced-Pt atoms by the formic acid via Pt-N bonding that is sandwiched in between two formic acid dimer planes. Figure 6 describes the most stable structure of the Pt-HMT complex for the initialization of 2D crystal growth of Pt in this process. It is true that the formic acid dimer plane on the top of the tetrahedral complex of 4Pt-HMT might be not as stable as the one at the basal plane of the tetrahedral structure with 3Pt-HMT at the beginning of their formation. However, with the increase of the concentration of the 4Pt-HMT assembly, the formic acid dimer plane at the top of the Pt-HMT complex becomes stable and can function as a scaffold for 2D crystal growth of Pt. Then, under a highly kinetic process in the presence of a strong reducing agent of formic acid, the Pt atoms crystalize to form 2D nanocrystals. 

Nevertheless, the exact mechanism of the 2D crystal growth of Pt requires a detailed step-by-step analysis during the growth process. Owing to the growth process being very rapid, special spectroscopic analysis is necessary to understand its evolution and the intermediate structure during the growth. We are in the process of acquiring this analysis and will present it in a different report.

## 3. Methods and Materials

2D crystal of Pt was prepared using a simple aqueous phase wet-chemical process at a mild temperature reaction. In a typical procedure, 5.0 M formic acid (Fluka, St. Louis, MO, USA) was added into a 15 mL aqueous solution containing 10 mM hexamethylenetetramine (Sigma-Aldrich, St. Louis, MO, USA) and 2 mM potassium hexa-chloro-platinate (K_2_PtCl_6_, Fluka). The solution was continuously stirred at 400 rpm during the growth process at 40 °C. The growth time was from 1 to 14 h. Black precipitate was obtained after the completion of the reaction and purified via centrifugation four times at 2000 rpm for 15 min. The purified precipitate was then re-dispersed into pure water. All chemicals were used as received without any further purification process. Pure water (~18 MΩ) was used throughout the reaction, which was obtained from a Milli-Q water purification system. 

The morphology of the sample was examined using a Transmission Electron Microscope (S/TEM) FEI Tecnai G2 F20. The instrument is also equipped with an Energy-Dispersive X-ray (EDX) apparatus with X-max^N^ 80T detector (Oxford Instrument, Abingdon, UK) for elemental analysis. The Selected-Area Electron Diffraction (SAED) analysis was carried out using a Hitachi HT7700 High-Contrast Digital TEM apparatus with 120 kV acceleration voltage. The lateral dimension of the 2D crystal of Pt was also verified using field-emission electron microscopy analysis by Carl Zeiss Supra 55VP FESEM system. Meanwhile, the thickness of the sample was evaluated using atomic force microscopy analysis by Ntegra Prima Atomic Force Microscopy system operated under the tapping modes operation. To verify its crystalline phase, we carried out low-angle X-ray diffraction spectroscopy using a Bruker D8 Advance XRD spectrometer with an incident X-ray grazing angle of 1.8° from horizontal. For the XRD analysis, the samples were dropped onto a clean Si substrate. To avoid strong interference from the Si diffraction peaks, the diffraction angle scanning was limited to the range of 2θ of 30 to 40°.

## 4. Conclusions

In conclusion, an ultimate thin 2D crystal in Pt can be realized in the aqueous phase via a reduction of the ions using formic acid in the presence of HMT. The thickness of the 2D crystal of Pt is 1.55 nm. This ultra-thin 2D crystal Pt may find potential application in the currently existing application and may provide a new dimension of research for scientific and industrial applications of these materials. 

## Figures and Tables

**Figure 1 ijms-23-10239-f001:**
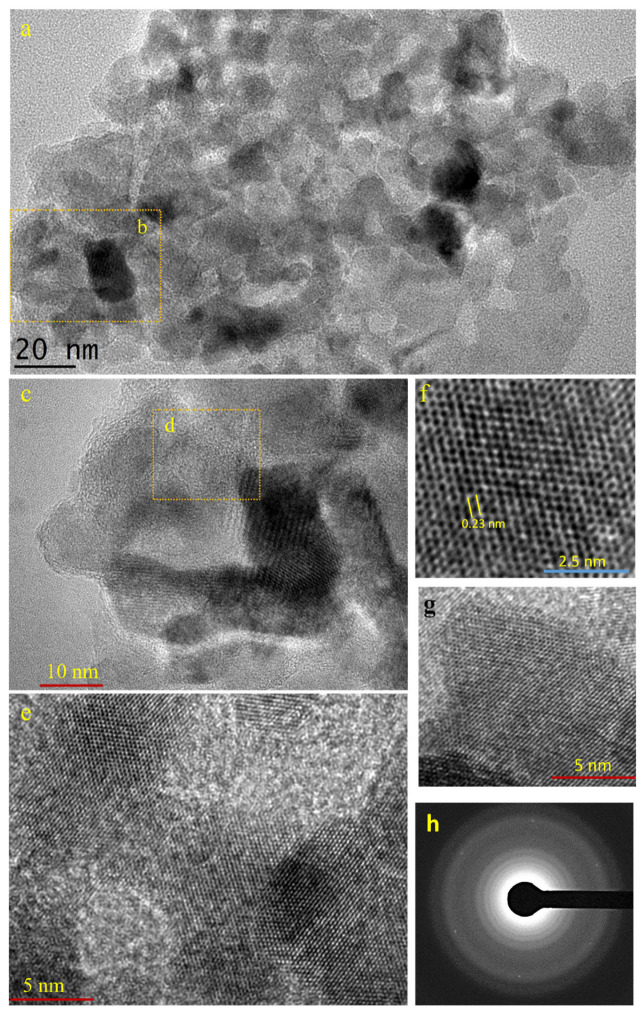
TEM image of 2D crystal of Pt. (**a**), Low-resolution transmission microscopy image of 2D crystal of Pt. (**b**), The location for higher resolution analysis in (**c**). (**d**), The location for higher resolution analysis in (**e**). (**e**–**g**), The typical higher-resolution transmission microscopy image of the 2D crystal of Pt sample. Image in (**f**) shows the lattice resemble graphene-like structure. (**h**), Selected-area electron diffraction (SAED) profile of 2D crystal of Pt on a carbon film-coated copper grid. The halos come from the amorphous carbon film.

**Figure 2 ijms-23-10239-f002:**
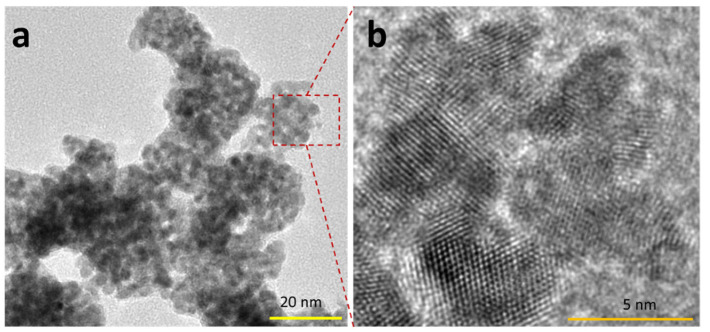
Thin and irregular shape nanostructure product. Low (**a**) and high (**b**) resolution images.

**Figure 3 ijms-23-10239-f003:**
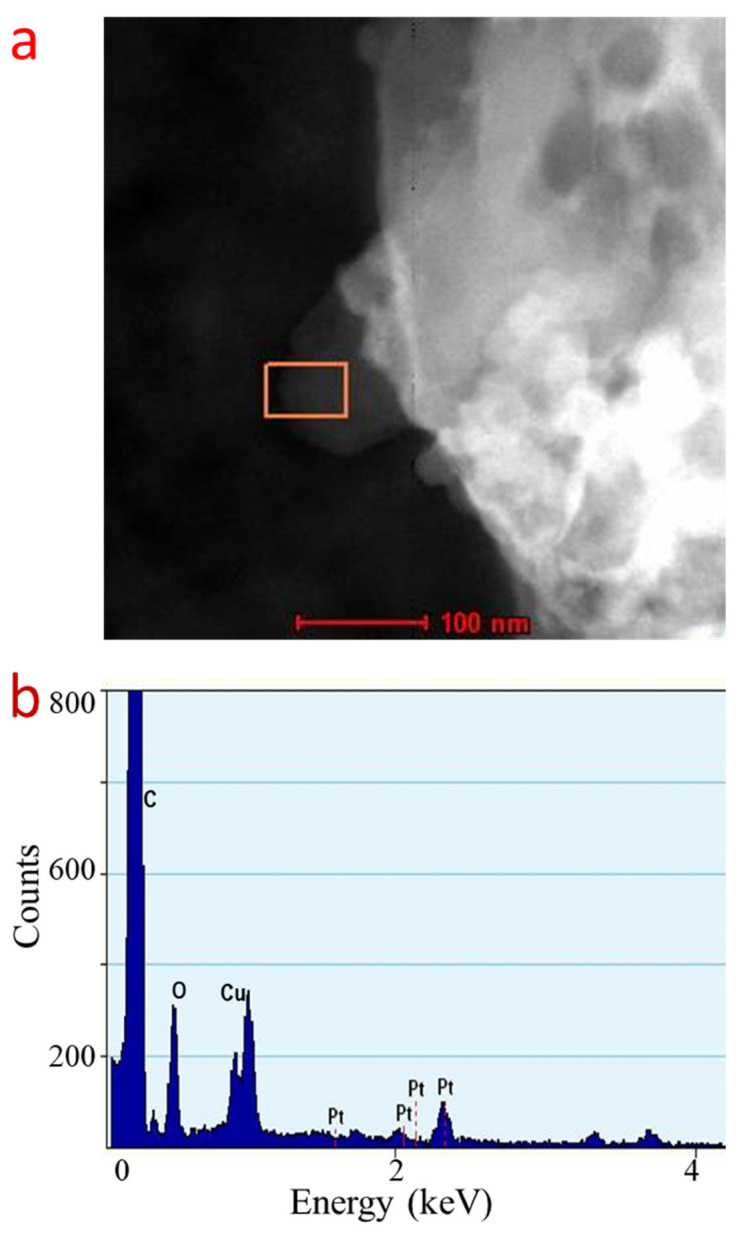
Elemental analysis of 2D crystal of Pt. (**a**), Electron image of the 2D crystal of Pt. (**b**), EDS elemental spectroscopy at the position shown as box in (**a**).

**Figure 4 ijms-23-10239-f004:**
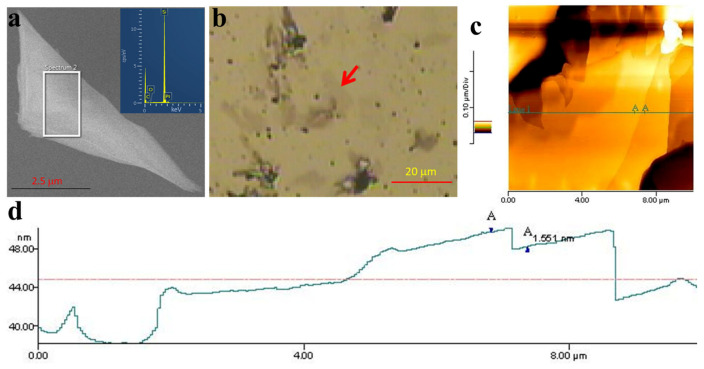
Dimension and thickness analysis of the 2D crystal of Pt. (**a**) The FESEM image of the 2D crystal of Pt and its elemental analysis result (inset). (**b**) The confocal microscope image of the 2D crystal of Pt on the substrate surface. The red arrow in (**b**) shows the ultra-thin Pt nanosheet structure. The red arrow also indicates the position used for AFM analysis results. (**c**,**d**) are the AFM image and the line profile analysis result of the 2D crystal of Pt. The sample was dispersed on the Si substrate.

**Figure 5 ijms-23-10239-f005:**
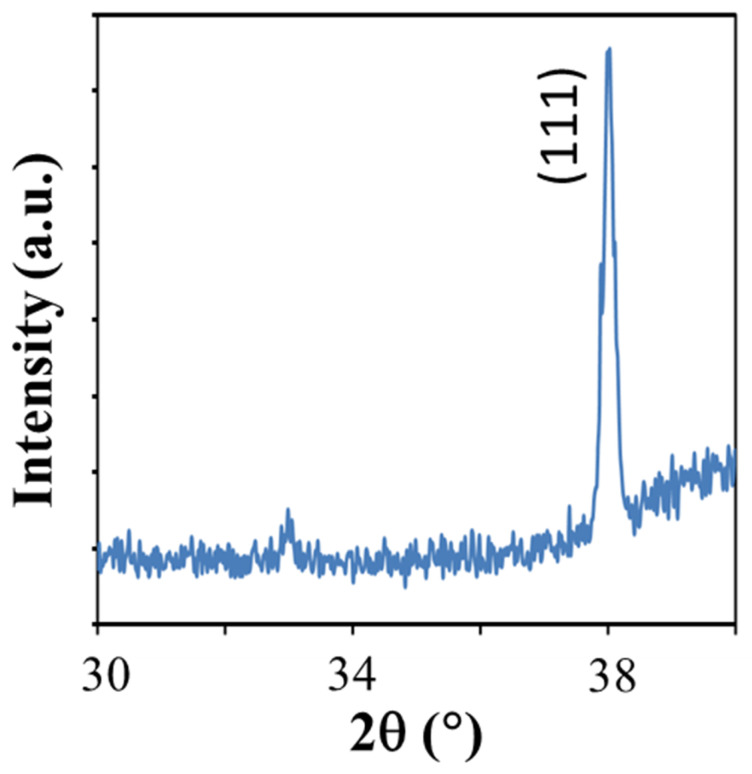
Grazing-angle X-ray diffraction spectra of 2D crystal of Pt.

**Figure 6 ijms-23-10239-f006:**
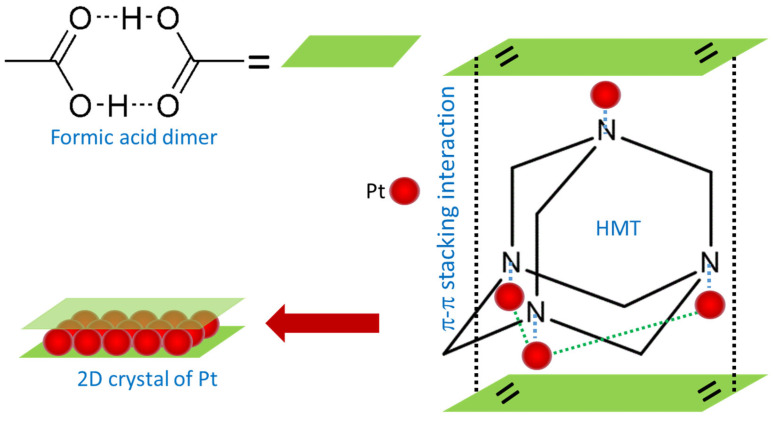
Mechanism of ultimate 2D crystal of Pt. Reduced Pt ion is predicted to coordinate with N at the corner of hexamethyletetramine (HMT) molecule. The formic acid dimer planes may then cover up the metal-attached HMT via π-π stacking interaction to form a 2D scaffold.

## Data Availability

The data are available from the authors upon request.

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
