# Peer review of "Formation of Ultimate Thin 2D Crystal of Pt in the Presence of Hexamethylenetetramine"

_ijms, 2022, doi:10.3390/ijms231810239_

Round 1
Reviewer 1 Report
The article is devoted to the production of ultimate thin platinum crystals when precipitated from an aqueous solution of potassium hexachloroplatinate in reaction with formic acid in the presence of urotropin. The established method for obtaining platinum crystals deserves attention, since the proposed method provides conditions for 2D anisotropic crystal growth in the orientation of the plane (111), which is a remarkable effect for platinum, given the cubic symmetry of its crystal structure.
There are the following comments to the presented work.
1. There are works in the literature on the study of the growth of Pt nanocrystals of various shapes (for example, [J. Phys. Chem. B 2005, 109, 188-193 (DOI: 10.1021/jp0464775)], [Appl. Organometal. Chem. 2006; 20: 638 – 647 (https://doi.org/10.1002/aoc.1123)], in which the structure and morphology of the crystals were studied by SEM and TEM methods. The authors could refer to similar works in the introduction in order to compare the results obtained.
2. The morphology of a crystal is its appearance, and not the projection during transmission, as in this case. Therefore, in the caption to the Fig. 1, it would be worth writing “TEM images of 2D crystal of Pt”.
3. Judging by the image in Fig. 1a, the crystal shown is a polycrystalline aggregate with a non-uniform thickness. The remark that "... that the 2D crystal is a perfect ultra-thin sheet with a very smooth and flat structure" (lines 88-89) is valid only for a selected local area for one of the single-crystal domains included in the polycrystal shown on Fig. 1a.
4. It is unclear how the authors estimated the yield share of 2D Pt crystals (40%). It would be worth writing about this in more detail.
5. Confocal image in Fig.4b is unsuccessful, it is of poor quality and it is difficult to find out anything from it. A satisfactory image of thin films with a thickness of several nm on the substrate surface can be obtained under phase contrast conditions using DIC Nomarski's.
6. To the AFM image in Fig.4c also has questions. This topographic image is of unsatisfactory quality for drawing convincing conclusions from it regarding the surface micro-morphology of the obtained crystals and their thickness. As an example, I can recommend for consideration works in which ultrathin (consisting of one or more monolayers) organic crystals were studied on substrates, including by the AFM method: Nature Comm. (2019) 10(1) (DOI: 10.1038/s41467-019-08573-8); ACS Appl. Mater. Interfaces 2019, 11, 6, 6315–6324 (DOI: 10.1021/acsami.8b20700). On the presented horizontal cross-section profile, the authors found a step with a height of 1.55nm, from which a conclusion was made about the thickness of the crystal (the designations and numbers on the cross-section profile scale in Fig.4d are poorly readable). It may be so, judging by the fact that crystals, as the authors write, tend to stack onto each other on the substrate for the sake of minimizing surface energy, but this cannot be proved from the topographic image presented, since this step may simply be a step of growth on the surface of a developed crystal face. If the resulting suspension with flat Pt crystals were to be processed in an ultrasonic bath, then the crystalline conglomerates could be divided into separate individuals, and then by deposition on a substrate, it is likely to study them more thoroughly both in terms of thickness and surface micro-morphology. Clear data on the surface micro-morphology of crystals with the involvement of SEM could shed light on the mechanism of their growth.
7. There is no description of sample preparation for low-angle X-ray diffraction analysis. Fig. 5 shows a part of the obtained spectrum for a peak at 38 degrees, but what is the full picture?
8. Now let's move on to the proposed scheme in Fig. 5, explaining the mechanism of formation of flat crystals Pt. In my opinion, one could give an equation for the chemical reaction of hexachloroplatinate with formic acid to reduce platinum, or at least discuss this point in a few words. The presence of urotropin (HMT) also triggers some chemical reactions. Unfortunately, the presented model is poorly substantiated and speculative. Formic acid molecules in the medium can and probably form dimers among themselves. However, the fact that these dimers can be assembled into stable aggregates due to pi-pi stacking interaction, especially when a tetrahedral complex of urotropin with Pt ions is wedged between them is a big question that requires serious justification. The only thing we can agree with is that specific complexation in the medium creates conditions for 2D anisotropy of Pt crystal growth in the directions of the plane (111). To clarify the mechanism of formation and growth of flat platinum crystals, detailed studies of the surface and shape of crystals by SEM and AFM methods could help.
9. Typo in line 152: plans -> planes.

Author Response
29 August 2022
Dear Editor,
We would like to say thank you very much for your email dated 22 August 2022, regarding the revision of our manuscript (Manuscript ID: ijms-1884993) submitted to this journal. We very much appreciate that you have given us the opportunity to revise this manuscript. We also would like to thank you very much to the reviewers for giving us a very valuable comments and suggestions. Based on those comments we try to improve the manuscript. Thus, we would like to re-submit the revised manuscript for your further consideration.
In this version, we have fully considered all comments and suggestions from the reviewers and we have made change in the manuscript based on that. We show the changes in the paper by the reddened and underlined text. We use “Track Change mode” tool (MS Word) for this purpose. In the following, we list down our responses to the comments of the reviewers.
Comment Reviewer 1.
The article is devoted to the production of ultimate thin platinum crystals when precipitated from an aqueous solution of potassium hexachloroplatinate in reaction with formic acid in the presence of urotropin. The established method for obtaining platinum crystals deserves attention, since the proposed method provides conditions for 2D anisotropic crystal growth in the orientation of the plane (111), which is a remarkable effect for platinum, given the cubic symmetry of its crystal structure.
Reply: We appreciate very much the reviewer’s comment on the mechanism of 2D crystal of Pt formation we proposed in this manuscript. We discuss the point and additional information is presented in this revision to support the proposed mechanism.
There are the following comments to the presented work.
- There are works in the literature on the study of the growth of Pt nanocrystals of various shapes (for example, [J. Phys. Chem. B 2005, 109, 188-193 (DOI: 10.1021/jp0464775)], [Appl. Organometal. Chem. 2006; 20: 638 – 647 (https://doi.org/10.1002/aoc.1123)], in which the structure and morphology of the crystals were studied by SEM and TEM methods. The authors could refer to similar works in the introduction in order to compare the results obtained.
Reply: We review the suggested references and discuss and cite them in the introduction section as ref 17 and ref 22.
- The morphology of a crystal is its appearance, and not the projection during transmission, as in this case. Therefore, in the caption to the Fig. 1, it would be worth writing “TEM images of 2D crystal of Pt”.
Reply: Thank you very much for the reviewer suggestion. We revise the caption of Figure 1 according to the reviewer’s suggestion.
- Judging by the image in Fig. 1a, the crystal shown is a polycrystalline aggregate with a non-uniform thickness. The remark that "... that the 2D crystal is a perfect ultra-thin sheet with a very smooth and flat structure" (lines 88-89) is valid only for a selected local area for one of the single-crystal domains included in the polycrystal shown on Fig. 1a.
Reply: thank you for the referee suggestion, we revise the phrase in page 4 to the following: “This, therefore, reveals that the localized 2D crystal is a perfect ultra-thin sheet with a very smooth and flat structure.”
- It is unclear how the authors estimated the yield share of 2D Pt crystals (40%). It would be worth writing about this in more detail.
Reply: We estimated the yield of the 2D crystal of Pt from the TEM image of size 200 nm x 200 nm. We add this explanation in the manuscript in page 4.
- Confocal image in Fig.4b is unsuccessful, it is of poor quality and it is difficult to find out anything from it. A satisfactory image of thin films with a thickness of several nm on the substrate surface can be obtained under phase contrast conditions using DIC Nomarski's.
Reply: we agree with the reviewer that the present confocal image may not reveal details of the structure. Unfortunately, our confocal microscope instrument has no DIC Nomarski option to enhance the contrast. However, the structure can be recognized from the image. We hope the present result can be considered.
We address this point in page 8 as follow: “…This is an extremely thin Pt nanostructure that may produce peculiar physicochemical properties for highperformance in existing applications. As shown in the inset of Figure 4d, the edge of the stacked 2D crystal could likely be tilted up. Despite this fact, the thickness of the layer can be roughly estimated from the result. However, a more detailed morphology analysis might be obtained via differential interference contrast (DIC) microscopy or Nomaski’s microscopy[20, 21]. Although Nomaski’s microscopy analysis is not available at present, the current result more or less verifies the existence of the ultimate thin 2D crystal of Pt.”
- To the AFM image in Fig.4c also has questions. This topographic image is of unsatisfactory quality for drawing convincing conclusions from it regarding the surface micro-morphology of the obtained crystals and their thickness. As an example, I can recommend for consideration works in which ultrathin (consisting of one or more monolayers) organic crystals were studied on substrates, including by the AFM method: Nature Comm. (2019) 10(1) (DOI: 10.1038/s41467-019-08573-8); ACS Appl. Mater. Interfaces 2019, 11, 6, 6315–6324 (DOI: 10.1021/acsami.8b20700). On the presented horizontal cross-section profile, the authors found a step with a height of 1.55nm, from which a conclusion was made about the thickness of the crystal (the designations and numbers on the cross-section profile scale in Fig.4d are poorly readable). It may be so, judging by the fact that crystals, as the authors write, tend to stack onto each other on the substrate for the sake of minimizing surface energy, but this cannot be proved from the topographic image presented, since this step may simply be a step of growth on the surface of a developed crystal face. If the resulting suspension with flat Pt crystals were to be processed in an ultrasonic bath, then the crystalline conglomerates could be divided into separate individuals, and then by deposition on a substrate, it is likely to study them more thoroughly both in terms of thickness and surface micro-morphology. Clear data on the surface micro-morphology of crystals with the involvement of SEM could shed light on the mechanism of their growth.
Reply: Thank you again for the reviewer point on the microscopy analysis result in Figure 4. We reviewed and cited the suggested literatures on the analysis of ultrathin organic film by AFM method. Based on the reference, we improved the presentation of thickness estimation in Figure 4. We also add an inset to the Figure 4d showing the clearer scale of the step.
Actually, the Pt crystals were grown in solution and dropped onto the Si surface to evaluate their morphology and thickness. We tought that the step could be resulted from the stacking of the 2D crystal during the solvent evaporation. Therefore, no further growth on the surface. Considering the presented line profiling result, the edge of the stacked crystal could be tilted from the horizontal plane. However, we could estimate a rough thickness of the structure.
We agree with the reviewer that the SEM analysis may help verify the real structure of the sample. However, we thought that the TEM result in Figure 1 would be sufficient to verify the existence of the 2D crystal of Pt. We hope the present result could be accepted for highlighting the existence of the thin structure of the Pt. We are in the process of detailing the real structure and properties of the 2D crystal of Pt and will report in different manuscript.
- There is no description of sample preparation for low-angle X-ray diffraction analysis. Fig. 5 shows a part of the obtained spectrum for a peak at 38 degrees, but what is the full picture?
Reply: We add the description of the low-angle XRD technique in our analysis. We used grazing angle of 1.8 degree from surface. The sample were droped on Si surface for this purpose. Broad angle scan did not show the Pt diffraction peaks because of strong Si diffraction background. Thus we scanned the reflection around the (111) diffraction peak, which is the strongest diffraction peak of fcc crystal, i.e. in the range of 2 theta of 30 and 40 degree. It is meant that the result is our complete diffraction data.
We address the point in page 3 as follows : “To verify its crystalline phase, we carried out low-angle X-ray diffraction spectroscopy using Bruker D8 Advance XRD spectrometer with an incident X-ray grazing angle of 1.8° from horizontal. For the XRD analysis, the samples were dropped onto a clean Si substrate. To avoid strong interference from the Si diffraction peaks, the diffraction angle scanning was limited to the range of 2q of 30 to 40°.”
- Now let's move on to the proposed scheme in Fig. 5, explaining the mechanism of formation of flat crystals Pt. In my opinion, one could give an equation for the chemical reaction of hexachloroplatinate with formic acid to reduce platinum, or at least discuss this point in a few words. The presence of urotropin (HMT) also triggers some chemical reactions. Unfortunately, the presented model is poorly substantiated and speculative. Formic acid molecules in the medium can and probably form dimers among themselves. However, the fact that these dimers can be assembled into stable aggregates due to pi-pi stacking interaction, especially when a tetrahedral complex of urotropin with Pt ions is wedged between them is a big question that requires serious justification. The only thing we can agree with is that specific complexation in the medium creates conditions for 2D anisotropy of Pt crystal growth in the directions of the plane (111). To clarify the mechanism of formation and growth of flat platinum crystals, detailed studies of the surface and shape of crystals by SEM and AFM methods could help.
Reply: We agree with the reviewer that the proposed mechanism might be further verified via detailed step by step analysis during the growth reaction to make sure what happened prior to the 2D crystallization of the Pt. The proposed mechanism was based on the unique coordination of HMT or urotropin with a large range of atoms that are bound via metal-N bonding formation. Thus, we thought that the reduced Pt atom might be at the beginning bonded to the HMT before crystallization and anisotropic growth with the help of a 2D formic acid dimmer direction.
We address this point in page 10 as follow: “Nevertheless, the exact mechanism of the 2D crystal growth of Pt requires a detailed step-by-step analysis during the growth process. Owing to the growth process being very rapid, special spectroscopic analysis is necessary to understand the evolution of the growth process and the intermediate structure during growth process. We are in the process of acquiring the analysis and will present the analysis in a different report.”.
As presented in the literature, during the reaction of K2PtCl6 with formic acid (formula is provided), the Pt ion will be reduced by the formic acid to form a Pt atom along with CO2 gas, KCl and H2 gas. In the presence of HMT, the Pt ion might be temporarily bonded to N atom of HMT. It is true that the HMT might also take part in the reaction. Considering the unique role of HMT as surfactant in wide range of nanoscrystal growth, we thought the HMT might not actively react with other reagent in the reaction, modifying its original properties. Combinatorial effect of formic acid dimers scaffold and HMT 2D crystal of Pt is produced.
Finally, despite the point-by-points raised by the reviewers being critical, we have put a serious attempt to carefully address the issues and provided additional information and discussion to clarify the problem. Thus, we hope the revised paper can be considered for publication in this journal.
With my best regards
Akrajas Ali Umar

Reviewer 2 Report
The scientific content of the ms. is referring to the production of ultimate thin 2D crystal of Pt which is implemented by a combinative effect of formic acid reductant and hexamethylenetetramine surfactant during the reduction of their metal ions precursor. The present study is well-written and with important scientific impact. Two minor points that the authors should consider by the authors before this article is accepted are the following:
(a) Introduction: The main novelty of this work is that the authors have used “a simple aqueous phase reduction of Pt precursors”. However, this novelty, in order to be highlighted, a better introduction on existing state of the art methods is crucial. The state of the art concerning existing synthetic approaches for ultra thin 2D crystals could be improved highlighting their limitations in order to reveal the significance of the present method. For instance, see Chin, HT., Hofmann, M., Huang, SY. et al. Ultra-thin 2D transition metal monochalcogenide crystals by planarized reactions. npj 2D Mater Appl 5, 28 (2021). https://doi.org/10.1038/s41699-021-00207-2.
(b) References: The number of references is limited. More recent research work on the field could be added.
Author Response
29 August 2022
Dear Editor,
We would like to say thank you very much for your email dated 22 August 2022, regarding the revision of our manuscript (Manuscript ID: ijms-1884993) submitted to this journal. We very much appreciate that you have given us the opportunity to revise this manuscript. We also would like to thank you very much to the reviewers for giving us a very valuable comments and suggestions. Based on those comments we try to improve the manuscript. Thus, we would like to re-submit the revised manuscript for your further consideration.
In this version, we have fully considered all comments and suggestions from the reviewers and we have made change in the manuscript based on that. We show the changes in the paper by the reddened and underlined text. We use “Track Change mode” tool (MS Word) for this purpose. In the following, we list down our responses to the comments of the reviewers.
Comment from Reviewer 2.
The scientific content of the ms. is referring to the production of ultimate thin 2D crystal of Pt which is implemented by a combinative effect of formic acid reductant and hexamethylenetetramine surfactant during the reduction of their metal ions precursor. The present study is well-written and with important scientific impact. Two minor points that the authors should consider by the authors before this article is accepted are the following:
Reply: We thank you very much for the reviewer encouragement and support, we in the following address the reviewer concerns.
(a) Introduction: The main novelty of this work is that the authors have used “a simple aqueous phase reduction of Pt precursors”. However, this novelty, in order to be highlighted, a better introduction on existing state of the art methods is crucial. The state of the art concerning existing synthetic approaches for ultra thin 2D crystals could be improved highlighting their limitations in order to reveal the significance of the present method. For instance, see Chin, HT., Hofmann, M., Huang, SY. et al. Ultra-thin 2D transition metal monochalcogenide crystals by planarized reactions. npj 2D Mater Appl 5, 28 (2021). https://doi.org/10.1038/s41699-021-00207-2.
Reply: Thank you again to the reviewer. We have reviewed the suggested reference and cited it accordingly due to its close relationship to the discussion presented in the paper. The ref is labeled as Ref 1.
We have also added an additional literature review on the state of the art of the 2D Pt synthesis, which is highlighted in the Introduction section.
(b) References: The number of references is limited. More recent research work on the field could be added.
Reply: During the preparation of the revision, we have added a number of new references, covering the technique and the characterization of Pt nanoplates.
Finally, despite the point-by-points raised by the reviewers being critical, we have put a serious attempt to carefully address the issues and provided additional information and discussion to clarify the problem. Thus, we hope the revised paper can be considered for publication in this journal.
With my best regards
Akrajas Ali Umar

Round 2
Reviewer 1 Report
The presented revision of the article is a significantly improved version. For the detailed characterization of ultrathin (in the limit of monolayer) crystals of such a remarkable material as platinum, it is still necessary to involve additional appropriate methods of structural diagnostics, which would allow to obtain more information about the mechanism of formation of these objects, and therefore at the current stage it is reasonable to use more restrained formulations.
Still, there are a few comments.
1. A small note to Fig. 4b: please specify the scale.
2. (Line 133) The slope of the crystal step relative to the horizontal axis on the graph of Fig. 4d, which the authors are talking about, is such that the difference is 2 nm vertically by 2 microns horizontally. This is a tilt of 0.05 degrees, and what it can be connected with, cannot be said. It is possible that it is associated with the inclination of the substrate relative to the microscope, since it is impossible to fix the sample with such high accuracy. Therefore, before publishing such data, the alignment procedure is applied by subtracting a plane or surface of a greater order using AFM-data editors (for example, free Gwyddion). In order to talk about the crystal inclination regarding the substrate, it is necessary that the region be visible in the image, which could be with some degree of confidence to say that this is a substrate. In the presented data, the substrate is not observed, so it is impossible to discuss the slope of the crystals relative to it.
3. As for the proposed model. As a hypothesis, it has the right to exist. I can only note that the 4Pt*HMT complex has tetrahedral coordination and symmetry, in connection with which, presented in Fig. 5 the configuration with possible planes of formic acid dimers can be somehow stable only from the side of the lower face of the tetrahedron formed by three Pt atoms, and the position at the top is unstable. For further development of the model, it may be worth considering the construction of tetrahedral complexes 4Pt* HMT, which are tightly packed into flat structures for which excess formic acid molecules (and also water is present!) they performed the role of some solvate shells limiting the assembly space. These considerations are only wishes and do not oblige the authors to respond to them in this work.
In general, the presented work is relevant and of interest for the development of the theory and practice of crystallization of low-dimensional systems, and I recommend it after minor edits for publication in IJMS.

Author Response
Reviewer comment.
The presented revision of the article is a significantly improved version. For the detailed characterization of ultrathin (in the limit of monolayer) crystals of such a remarkable material as platinum, it is still necessary to involve additional appropriate methods of structural diagnostics, which would allow to obtain more information about the mechanism of formation of these objects, and therefore at the current stage it is reasonable to use more restrained formulations.
Reply: Thank you again for the reviewer point regarding the formation mechanism of the 2D crystal of Pt. We agree that detailed analysis is necessary to understand the growth mechanism. We are attempting the study and trying an appropriate techniques to figure out the growth process. We hope the current hypothesis for the growth mechanism can be considered. We address the suggestion the reviewer wrote in the manuscript. And also we very much appreciate for the reviewer recommendation and support for the publication of our manuscript in this journal.
Still, there are a few comments.
- A small note to Fig. 4b: please specify the scale.
Reply: We add the scale to the Figure 4b.
- (Line 133) The slope of the crystal step relative to the horizontal axis on the graph of Fig. 4d, which the authors are talking about, is such that the difference is 2 nm vertically by 2 microns horizontally. This is a tilt of 0.05 degrees, and what it can be connected with, cannot be said. It is possible that it is associated with the inclination of the substrate relative to the microscope, since it is impossible to fix the sample with such high accuracy. Therefore, before publishing such data, the alignment procedure is applied by subtracting a plane or surface of a greater order using AFM-data editors (for example, free Gwyddion). In order to talk about the crystal inclination regarding the substrate, it is necessary that the region be visible in the image, which could be with some degree of confidence to say that this is a substrate. In the presented data, the substrate is not observed, so it is impossible to discuss the slope of the crystals relative to it.
Reply: We agree with the reviewer that an appropriate alignment procedure should be conducting on the AFM data. Actually we have applied the substrate alignment process on the AFM data prior to the scanning process. However, the AFM data was not processed for possible improving the structure dimensionality, such as using Gwyddion (we are not familiar with this software and with evaluate its functionalities). Therefore, we hope the present data could be considered to roughly describe the presence of the 2D crystal of Pt.
We also realize that we overlooked to include the substrate background during AFM scanning. This was due to the attempt to increase the resolution of the scanning as the scanning conducted on the surface of the nanosheet structure. By not including the substrate background we could obtained the layered structure of the 2D crystal of Pt. Nevertheless, we take a note on the reviewer point and suggestion and will carry out more detailed analysis on the structure and the growth mechanism.
We address the reviewer comment on this point in page 8 as follow:” As shown in the inset of Figure 4d, the edge of the stacked 2D crystal could likely be tilted up. Nevertheless, the definite structure and dimensionality of the 2D crystal of Pt might be further obtained by applying a proper processing technique of the AFM data. This includes the elimination of the substrate inclination relative to the AFM probe so the accuracy of the layered structure thickness or spacing can be increased. Despite this fact, the thickness of the layer can be roughly estimated from the result. In addition, a more detailed confocal microscope image of the 2D crystal of Pt might also be obtained via differential interference contrast (DIC) microscopy or Nomaski’s microscopy[25, 26]. However, although Nomaski’s microscopy analysis is not available at present, the current result more or less verifies the existence of the ultimate thin 2D crystal of Pt.”
- As for the proposed model. As a hypothesis, it has the right to exist. I can only note that the 4Pt*HMT complex has tetrahedral coordination and symmetry, in connection with which, presented in Fig. 5 the configuration with possible planes of formic acid dimers can be somehow stable only from the side of the lower face of the tetrahedron formed by three Pt atoms, and the position at the top is unstable. For further development of the model, it may be worth considering the construction of tetrahedral complexes 4Pt* HMT, which are tightly packed into flat structures for which excess formic acid molecules (and also water is present!) they performed the role of some solvate shells limiting the assembly space. These considerations are only wishes and do not oblige the authors to respond to them in this work.
Reply: We also agree with the reviewer point that the proposed mechanism indicates that the formic acid dimer plane at the top structure of the Pt-HMT complex will be not stable. However, we thought that with the increase of Pt-HMT complex assembly, the top dimeric plane become stable and can function as scaffold for 2D crystal growth of Pt.
We address this point in the manuscript at page 10 as follow:” Figure 6 describes the most stable structure of the Pt-HMT complex for the initialization of 2D crystal growth of Pt in this process. It is true that the formic acid dimer plane on the top of the tetrahedral complex of 4Pt-HMT might be not as stable as the one at the basal plane of the tetrahedral structure with 3Pt-HMT at the beginning of their formation. However, with the increase of the concentration of the 4Pt-HMT assembly, the formic acid dimer plane at the top of the Pt-HMT complex become stable and can function as scaffold for 2D crystal growth of Pt. Then, under a highly kinetic process in the presence of a strong reducing agent of formic acid, the Pt atoms then crystalize to form 2D nanocrystals.”